# Frenetic, under-Challenged, and Worn-out Burnout Subtypes among Brazilian Primary Care Personnel: Validation of the Brazilian “Burnout Clinical Subtype Questionnaire” (BCSQ-36/BCSQ-12)

**DOI:** 10.3390/ijerph17031081

**Published:** 2020-02-08

**Authors:** Marcelo Demarzo, Javier García-Campayo, David Martínez-Rubio, Adrián Pérez-Aranda, Joao Luiz Miraglia, Marcio Sussumu Hirayama, Vera Morais Antonio de Salvo, Karen Cicuto, Maria Lucia Favarato, Vinicius Terra, Marcelo Batista de Oliveira, Mauro García-Toro, Marta Modrego-Alarcón, Jesús Montero-Marín

**Affiliations:** 1Mente Aberta-Brazilian Center for Mindfulness and Health Promotion, Department of Preventive Medicine, Universidade Federal de São Paulo (UNIFESP), 13565-905 São Paulo, Brazil; demarzo@unifesp.br (M.D.); joaomiraglia@gmail.com (J.L.M.); sussumu.hirayama@gmail.com (M.S.H.); desalvo@terra.com.br (V.M.A.d.S.); karencicuto@hotmail.com (K.C.); malufavarato@gmail.com (M.L.F.); vinicius.loyola@hotmail.com (V.T.); mbdo2@gmail.com (M.B.d.O.); 2Primary Care Prevention and Health Promotion Research Network (RedIAPP), 50009 Zaragoza, Spain; jgarcamp@gmail.com (J.G.-C.); martamodal@gmail.com (M.M.-A.); 3Miguel Servet Hospital and University of Zaragoza, RedIAPP, Instituto Aragonés de Ciencias de la Salud, 50009 Zaragoza, Spain; 4Excellence Research Network PROMOSAM (PSI2014-56303-REDT), 28029 Madrid, Spain; 5Psicoforma Integral Psychology Center, 46001 Valencia, Spain; 6Teaching, Research & Innovation Unit, Parc Sanitari Sant Joan de Déu, 08830 St. Boi de Llobregat, Spain; 7Faculty of Psychology, Universitat Autònoma de Barcelona, Bellaterra (Cerdanyola del Vallès), 08193 Barcelona, Spain; 8Department of Psychology, University of the Balearic Islands, 07122 Mallorca, Spain; mauro.garcia@uib.es; 9Department of Psychiatry, University of Oxford, Warneford Hospital, Oxford OX3 7JX, UK; jesus.monteromarin@psych.ox.ac.uk

**Keywords:** burnout, burnout subtypes, BCSQ, primary care, Brazil, validation studies, bifactor, questionnaire

## Abstract

Primary healthcare personnel show high levels of burnout. A new model of burnout has been developed to distinguish three subtypes: frenetic, under-challenged, and worn-out, which are characterized as overwhelmed, under-stimulated, and disengaged at work, respectively. The aim of this study was to assess the psychometric properties of the long/short Brazilian versions of the “Burnout Clinical Subtypes Questionnaire” (BCSQ-36/BCSQ-12) among Brazilian primary healthcare staff and its possible associations with other psychological health-related outcomes. An online cross-sectional study conducted among 407 Brazilian primary healthcare personnel was developed. Participants answered a Brazil-specific survey including the BCSQ-36/BCSQ-12, “Maslach Burnout Inventory-General Survey”, “Utrecht Work Engagement Scale”, “Hospital Anxiety/Depression Scale”, “Positive-Negative Affect Schedule”, and a Visual Analogue Scale of guilt at work. The bifactor was the model with the best fit to the data using the BCSQ-36, which allowed a general factor for each subtype. The three-correlated factors model fit better to the BCSQ-12. Internal consistence was appropriate, and the convergence between the long-short versions was high. The pattern of relationships between the burnout subtypes and the psychological outcomes suggested a progressive deterioration from the frenetic to the under-challenged and worn-out. In sum, the Brazilian BCSQ-36/BCSQ-12 showed appropriate psychometrics to be used in primary healthcare personnel.

## 1. Introduction

Burnout syndrome is one of the most important work-related conditions associated with chronic stress and mental and/or physical weariness. It has been said that around 40% of primary care (PC) health workers may suffer from burnout, what can lead to reduced productivity and quality of care and to increased absenteeism and health care costs [1,2,3,4,5].

Burnout has traditionally been defined by means of the dimensions of exhaustion, cynicism, and professional (lack of) efficacy, belonging to the “Maslach Burnout Inventory” (MBI) [6]. Exhaustion is the feeling of not being able to offer any more of oneself at work, as the consequence of a prolonged exposure to excessive demands. Cynicism is a detached attitude to tasks, colleagues, and recipients of service. Finally, inefficacy is a perception of not performing tasks adequately with feelings of incompetence. The MBI has been considered a gold standard measure of burnout, but there is still disagreement in the scientific literature about how to define and measure this syndrome. This standard conceptualization has presented flaws in its measurement format, such as the lack of psychometric consistency in its structure, as well as the problem that it was not developed through a systematic theorization of the syndrome but simply through factor grouping of a rather arbitrary set of items [7]. This proposal has also been criticized because it is not able to clarify the relationships between its components or about antecedents and consequences of burnout [8,9,10]. The weakness of this definition has also been observed by the wide range of prevalence that its use throws out [11]. In addition, there is a current trend that questions to what extent burnout is a psychiatric disorder or cluster on its own, proposing instead that burnout might “simply” be a specific dimension of major depressive disorders [12,13,14,15,16]. Thus, it has been noted that reliable conclusions can neither be drawn about burnout prevalence nor about its relationship with sociodemographic or clinical variables, and it has been suggested that improving the measurement instruments is crucial at this stage [11,17].

Other authors have highlighted that the burnout term is being widely used but poorly measured because the construct behind the classical definition could not be sufficiently valid due to a non-clinically based origin [17,18]. The latest revision of the International Classification of Diseases, ICD-11, introduces burnout as a work-related chronic state of stress and psychological exhaustion. However, it identifies this syndrome as an occupational phenomenon—not a medical condition—that is primarily related to the mismatch between the environment (i.e., demands) and the individual (i.e., resources that are necessary to develop the work meaningfully) [19]. One of the most significant downsides of the classical point of view of burnout, especially regarding the development of intervention strategies, is the fact that it evaluates all cases with a definition based on a scarce set of symptoms, whereas the psychosocial reality in which the syndrome develops tends to vary among cases [20,21]. In other words, burnout syndrome has usually been described as a uniform construct in all individuals, with a similar aetiology and group of symptoms [22]; however, experience in the treatment of this complex psychosocial entity suggests the need to identify different routes in the development of the syndrome in order to adjust for more effective lines of therapeutic action [21]. Given these limitations, other options are being explored to define and measure burnout [7,23,24]. In this line, as an alternative to the traditional definition, distinct burnout subtypes (e.g., frenetic, under-challenged, and worn-out) that point to various groups at risk of burnout syndrome have been proposed from Farber’s seminal clinical work [20,22,25,26,27,28,29,30]. The definitions of these burnout profiles are the result of a unifying methodological process, originating with this author’s clinical approach and based on a purely phenomenological description of cases under psychological treatment, which was subjected to a qualitative analysis of content prior to the subsequent validation, in psychometric terms, of the “Burnout Clinical Subtypes Questionnaire” (BCSQ-36/BCSQ-12) [31,32]. The BCSQ is a recently developed tool that permits the measurement of the referred three distinct burnout profiles. It allows the identification of risk groups rather than cases of burnout in a classical sense and, thus, highlights the complex and multifaceted nature of the syndrome by facilitating a more person-orientated approach to burnout [33]. This framework is focused on psychological processes that are relevant when intervening on the specific characteristics of each particular case [34]. The BCSQ has already been previously used in different languages and in several countries, such as Spain [31], Iran [35], Sri Lanka [36], India [37], and Austria [38], among others, and it is now being translated and validated in many others.

The frenetic burnout subtype describes individuals who work increasingly harder, seeking a success that is compatible with their efforts. They also demonstrate excessive ambition caused by their need to achieve important goals and are incapable of recognizing their own limitations and personal needs to the point of suffering from work overload. This subtype is associated with high levels of exhaustion and a coping style focused on active problem-solving—which is highly dysfunctional and actually does not usually lead to “solving the problem” when it is always used as the only coping strategy—with individuals employing a large number of working hours or getting involved in multiple tasks [25,31,32,39]. As a contrast, individuals described by the under-challenged subtype have to cope with monotonous and unstimulating conditions. They display indifference, performing their work-related tasks superficially without interest while experiencing boredom and a lack of development, which is understood as the absence of personal growth. This burnout subtype has been associated with high levels of cynicism and arises in jobs with bureaucratic and repetitive tasks. As a result, it presents an escapist coping style based on distraction and cognitive avoidance [25,31,32,39]. At first glance, the under-challenged subtype does not seem to match the key concept of burnout syndrome in classical terms, even more so if we consider burnout as a specific case of major depressive disorders. However, the previously indicated under-challenged profile seems to be also moderately related to exhaustion and lack of efficacy [31], creating a burnout risk group that might correspond to states of the syndrome that could have been overlooked in previous research [38]. The worn-out subtype describes workers who act carelessly when faced with stress and a lack of gratification. They usually present with feelings of hopelessness due to the lack of control over their work and what they perceive to be an absence of acknowledgement for the effort invested. Eventually, they opt for omission, disregard, and neglect as their preferential response to difficulties at work, and as a result, this subtype is strongly associated with the perception of a lack of efficacy and feelings of incompetence after years of service in organizations with unfair rewarding contingency systems [25,31,32,39]. This profile seems to best exemplify a fully established form of burnout. Nevertheless, it is important to highlight the fact that the three burnout subtypes have presented adequate discriminative values regarding general negative affectivity states using the Positive and Negative Affect Schedule (PANAS) questionnaire in PC healthcare personnel, with Pearson’s correlation coefficient values ranging from r = 0.24 to 0.29 [40].

The different burnout subtypes can be theoretically ordered into a continuum based on their level of dedication towards work-related tasks, from the frenetic subtype (highly involved) through the under-challenged (intermediate) until the worn-out (lowly dedicated) [25]. Variations in the levels of dedication appear to be the way in which individuals attempt to exert control over the psychological distress that leads to the perception of a lack of reciprocity in exchange relationships at work [28]. This feeling of imbalance between efforts invested and rewards obtained in social exchanges is an important source of distress, and its presence is associated with the manifestation of psychosomatic symptoms of burnout [41,42].

The Brazilian Unified National Health System (“Sistema Único de Saúde”) has a decentralized structure wherein municipalities are in charge of implementing PC, which is provided through the “Family Health Strategy”. The Family Health Strategy is a community-based approach aimed at covering important PC functions and is organized around family health teams, responsible for a geographically defined area with a population of up to 1000 households (approximately 3500 residents) each, with no overlap between areas. These teams are composed of a physician, a nurse, one or two nurse assistants, and four to six community health workers and volunteers. They operate at PC units with the support of a multidisciplinary team, known as “Family Health Supporting Team”, that may vary in its composition and can include dentists, nutritionists, social workers, psychologists, psychiatrists, physical education specialists, speech and hearing therapists, occupational therapists, and gynaecologist-obstetricians, among others [43,44].

The increase in PC coverage since the late nineties has been remarkable in Brazil, from zero in 1993 to about 64% of Brazilian population in 2019 (representing over 134 million people, mostly low income or underserved), and has marked a decrease in inequities of access that was translated into greater user satisfaction and a reduction in infant and adult mortality in addition to adult complications related to chronic noncommunicable diseases. However, in order to achieve universality, equity, and sustainability, the Family Health Strategy still faces many challenges that include chronic underfunding, heterogeneity of available physical resources and in the number and quality of health professionals, lack of integration into secondary and tertiary care, delay in the implementation of national electronic health records, and a growing ageing population [44,45,46]. These challenges should be of concern and may exert chronic stress among the PC provider workforce given that they have to deal with complex conditions such as mental health problems combined with economic and social issues [47]. In this regard, a number of studies have identified chronic stress and burnout syndrome as relevant issues among PC personnel in Brazil, highlighting their relation to the characteristics of the working environment [2,48,49,50] and prompting the need for further studies to provide better understanding of this condition, particularly with regard to real prevalence, risk factors, and efficacious interventions.

In this context and due to the absence of instruments to assess the burnout subtypes in Brazilian PC settings, this study was designed to culturally translate and evaluate the psychometric properties of the Brazilian version of the “Burnout Clinical Subtypes Questionnaire” in its long (BCSQ-36) and short (BCSQ-12) forms, with special emphasis in analysing the properties of the general common factor that underlies each burnout subtype—something that has not been done to date. We also aimed to explore the explanatory power of the burnout subtypes on the classical burnout dimensions of exhaustion, cynicism, and (lack of) efficacy and the possible relationships between the burnout subtypes and other important psychological health-related variables such as engagement, anxiety, depression, affectivity, and guilt at work. Finally, we tried to estimate possible sociodemographic and occupational factors that might be related to the burnout subtypes among Brazilian PC health professionals.

We hypothesized that (1) the BCSQ maintains adequate psychometrics in their Brazilian translation; (2) the BCSQ significantly explains the burnout symptoms according to the classical definition; (3) the frenetic subtype is mainly related to exhaustion, the under-challenged is mainly related to cynicism, and the worn-out is mainly related to (lack of) efficacy; (4) the other psychological health-related variables are related to the burnout subtypes in the same degree of their disengagement at work (from less to more: from the frenetic through the under-challenged to the worn-out); and (5) the burnout subtypes are differentially related to sociodemographic and occupational variables.

## 2. Materials and Methods

### 2.1. Design

This was a cross-sectional validation study conducted from May to July 2015. The study was revised and approved by the Ethic Committee of the “Universidade Federal de São Paulo” (UNIFESP; dated 26 October 2016; code: CAAE 40254314.8.0000.5505; this date is later than the date on which screening was conducted because subsequent additions were made to the project after its initial approval; however, those modifications did not affect this work). This study was conducted in accordance with the Declaration of Helsinki of 1975, revised in 2013. The data (totally anonymized) used to produce the study results and descriptions of the project are fully available on the following open access repository: http://doi.org/10.3886/E109282V1. More information such as the specific sequence interested researchers should take into account for replication studies as well as the computer codes used in the study will be posted on the corresponding group’s website (https://www.mindfulnessbrasil.com), and any other questions will be answered upon request.

### 2.2. Study Sample

PC health personnel affiliated to the Brazilian Society of Family and Community Medicine, registered at the Open University of the Unified National Health System of the Federal University of São Paulo (UNIFESP), or working at the city of Santos or Santo André (all of them in the state of São Paulo) configured the total universe of PC professionals, with 9367 listed in April 2015. The sample size was estimated to exceed the recommended 10:1 ratio for the number of subjects to the number of test items to ensure the psychometric adequacy of each burnout subtype scale, and it was therefore established that around 200 participants would be needed at minimum [51]. A low response rate was expected due to the online questionnaire type sent by email, so we inflated the target sample size to 1600 subjects randomly selected in a simple way from the total list of professionals to ensure that the final sample size would be adequate for the study. Following the signature of an online written informed consent, participants were invited to answer a self-reported online battery of questionnaires made available through the Survey Monkey platform (https://www.surveymonkey.com). The invitation to participate in the study was repeated 3 times in 3 weeks to achieve the greatest possible number of positive voluntary responses.

### 2.3. Translation/Back Translation Procedure

Two burnout experts independently performed the forward translation into Portuguese that was followed by back-translation by two bilingual linguistic experts with no prior knowledge of the BCSQ tool. Finally, the two back-translations were reviewed by a native Portuguese teacher, and any differences were resolved by mutual agreement among the translators. The usual guidelines for cross-cultural adaptations were followed [52,53,54]. The final Portuguese versions of the BCSQ-36/BCSQ-12 can be found in Appendix A, and the corresponding English version is shown in Appendix A.

### 2.4. Measures

#### 2.4.1. Sociodemographic and Occupational Characteristics

According to the previous work of Montero-Marin et al. [55], the following sociodemographic and occupational data and categorizations were obtained from participants: age, sex, relationship status (partnership/married or single), number of children (none, one, or more), category (volunteer or professional with a salary), job position (physician, nurse, or community health worker), hours worked per week (<40, 40, or >40), total length of service in years (<6, 6–16, or >16), years at the same job (<6, 6–16, or >16), contract period (temporary or permanent), contract type (full-time or part-time), economic difficulties (never, sometimes, almost always, or always), and sick leave days in the past year (yes—and number—or no).

#### 2.4.2. Burnout Clinical Subtypes Questionnaire (BCSQ-36/BCSQ-12)

The BCSQ in its long version (BCSQ-36) [31] is comprised of 36 items distributed into 3 scales, each of them with 3 subscales in turn. The frenetic scale includes involvement (e.g., “I get very involved in solving work-related problems”), ambition (e.g., “I am ambitious to obtain important results in my work”), and overload (e.g., “I think the dedication I invest in my work is more than what I should for my health”); the under-challenged scale includes indifference (e.g., “I feel indifferent about my work and have little desire to succeed”), lack of development (e.g., “My work does not offer me opportunities to develop my abilities”), and boredom (e.g., “I feel bored at work”); and the worn-out scale includes neglect (e.g., “I give up in the face of any difficulties in my work tasks”), lack of acknowledgement (e.g., “I think my dedication to my work is not acknowledged”), and lack of control (e.g., “I feel the results of my work are beyond my control”). On the other hand, the BCSQ-12 [41] is a short version of the BCSQ composed by 12 items from the dimensions of overload, lack of development, and neglect, which have shown the greatest convergence with the MBI gold standard and the most discriminant validity among the burnout subtypes [31]. Participants had to indicate their perceived agreement with each item following a Likert 7-point scale ranging between 1 (“totally disagree”) and 7 (“totally agree”). Scores are presented as a sum of its constituent items divided by the number of them (scalar scores). Higher scores mean greater levels of burnout. The psychometrics of the original version of the BCSQ have been adequate in previous studies [31,32,40].

#### 2.4.3. Maslach Burnout Inventory-General Survey (MBI-GS)

The “Maslach Burnout Inventory-General Survey” (MBI-GS) was used to measure the symptoms of burnout in their classical gold standard definition. We used the MBI-GS, a measure of burnout in general occupational groups other than human services work, in order to have comparable results with previous studies regarding the BCSQ [31,32,40,56]. The Brazilian version of the MBI-GS was validated by Tamayo and Tróccoli [57]. It consists of 16 items grouped into 3 dimensions: exhaustion (e.g., “I feel burned out from my work”; with ω = 0.93 in the present study), cynicism (e.g., “I have become less enthusiastic about my work”; ω = 0.72 in our study), and professional efficacy (e.g., “I feel confident that I am effective at getting things done”; ω = 0.82 in our study). The answers range between 0 (“never”) and 6 (“always”) following a Likert-type 7-point scale structure. Higher scores mean greater levels of burnout. The Portuguese version of the MBI-GS has shown adequate psychometric properties in its validation study [57].

#### 2.4.4. Utrecht Work Engagement Scale (UWES)

The “Utrecht Work Engagement Scale” (UWES) was used to evaluate the mental state of accomplishment at work, through 17 items grouped into the components of vigor (e.g., “At my work, I feel bursting with energy”; with ω = 0.89 in the present study), “dedication” (e.g., “I find the work that I do full of meaning and purpose”; with ω = 0.91 in our study), and “absorption” (e.g., “Time flies when I am working”; with ω = 0.86 in our study). Responses are arranged on a Likert-type scale that ranges between 0 (“never”) and 6 (“always”). Higher scores mean greater levels of engagement. The validated Brazilian version of the questionnaire was used [58].

#### 2.4.5. Hospital Anxiety and Depression Scale (HADS)

The “Hospital Anxiety and Depression Scale” (HADS) is used to assess possible cases of anxiety or depression among nonpsychiatric populations. It was developed for use in general medical outpatient clinics, but this scale is also widely used in clinical practice and research on both clinical and general populations with strong psychometric properties [59,60,61]. The HADS is divided into the anxiety subscale (HADS-A, with 7 items, e.g., “I feel tense or wound up”, with ω = 0.84 in the present study) and the depression subscale (HADS-D, with 7 items, e.g., “I feel as if I am slowed down”, with ω = 0.83 in the present study), with both HADS-A and HADS-D presenting adequate psychometric characteristics in their Portuguese validation [62].

#### 2.4.6. Positive and Negative Affect Schedule (PANAS)

The “Positive and Negative Affect Schedule” (PANAS) is a self-report instrument that measures positive and negative affectivity by presenting a list of 20 adjectives, 10 per subscale, which are rated on a Likert-type 5-point scale (e.g., positive: “inspired”, ω = 0.87 in our study; negative: “ashamed”, ω = 0.92 in our study). Present moment time instructions were used in the present study. Both positive and negative subscales have shown appropriate psychometric features in a previous Brazilian study [63].

#### 2.4.7. Visual Analogue Scale of Guilt at Work

The burnout syndrome has been associated with the presence of guilt feelings, and this variable might play an important role in the development and chronification of the syndrome [64]. The visual analogue scale of guilt at work was designed for exploring general aspects of guilt at work in the present study and represents feelings of blame as a result of lack of success at work, desires for job change, and lack of responsibility when doing tasks [55]. Participants had to place a mark in a thermometer-like scale showing their perceived level of guilt at work, ranging between 0 (“none”) and 10 (“much”), in relation to the previously referred contents as a whole. This type of visual analogue scale is frequently used in psychological research of occupational environments, with adequate psychometric properties [65].

### 2.5. Statistical Analysis

The sociodemographic data were described using means (SDs) and frequencies (percentages) according to their distribution. The BCSQ items’ behaviours were assessed using means (SDs) as well as skewness and kurtosis indices. The means of the inter-item Pearson’s correlation and item-rest Pearson’s correlations were also calculated for each factor.

The fit of the models was examined by using confirmatory factor analysis (CFA) to (I) a one-factor solution taken as a reference; (II) a three-correlated factors model according to the theoretical features of each burnout profile, and (III) a bifactor approximation to evaluate the adequacy of considering a total score as a common factor in each burnout subtype and to evaluate the features of this general factor and possible departures from unidimensionality. The Robust Maximum Likelihood (MLR) method from raw data entry, which is usually advised for Likert-type scales that include ≥5 or more answer categories and is robust to the non-normality distribution of the data [66], was used. From a general perspective, model assessment was based on the following goodness-of-fit indices: comparative fit index (CFI), Tucker-Lewis Index (TLI), the root mean square error of approximation (RMSEA), and the standardized root mean square residual (SRMR). Higher values of 0.90 and 0.95 for CFI and TLI indicate adequate and excellent fit indices, respectively [67]. RMSEA and SRMR indicate adequate fit when <0.08 and an excellent fit when <0.06 [67]. The chi-square test (χ2), although reported, was not a decisive index because it is very sensitive to sample size. Nevertheless, we used chi-square/degrees of freedom (χ2/df), which indicates a good fit when <5, and an excellent fit when <3 [68]. The Akaike Information Criterion (AIC) was used to perform model comparisons, being that a lower AIC value indicates a better fit. The proportion of explained variance was also examined. From an analytical perspective, standardized factor loadings (λ), uniqueness terms (δ), and standardized inter-factor correlations between latent factors (φ) were considered.

Construct replicability (H) was evaluated as the proportion of the factor variance that can be accounted for by its indicators, with reasonable values when ≥0.70 and good values when ≥0.80 [69]. Quality of factor score estimates were quantified by using the factor determinacy index (FDI). FDI is the correlation between the factor score estimates and the levels on the latent factors they estimate, and values of around 0.80 are adequate [70]. The omega (ω) index for the total scale and for each subscale (ωS), the omega hierarchical (ωH) and the omega hierarchical-subscale (ωHS), were calculated as reliability composites [71]. The average variance extracted (AVE) by the construct in relation to the variance due to measurement error was also estimated. AVE presents appropriate values of construct convergent validity when ≥0.50. There is discriminant validity among subfactors when AVE values are greater than the squared correlation between them [72]. The proportion of explained common variance attributable to the general factor (ECV) was calculated [69], as was the percentage of uncontaminated correlations (PUC), which indicates the proportion of correlations reflecting the general factor. If ECV and PUC are >0.70, the common variance can be regarded as essentially unidimensional, but it is also accepted if PUC < 0.80 but ECV > 0.60 and when ωH > 0.70 [69].

Subsequent analyses were carried out using BCSQ factorial scores. Pearson’s coefficients were used to evaluate the relationships among the general factors of the long BCSQ-36 (e.g., frenetic, under-challenged, and worn-out), among the subscales of the short BCSQ-12 (e.g., overload, lack of development, and neglect)—those relationships between the long and short versions were adjusted for correlated errors (adj-*r*) [73]—and between them and the psychological health-related variables referred above. The explanatory power of the BCSQ long and short versions in relation to the MBI gold standard was estimated by multiple linear regression models. It has been proposed that the burnout subtypes might be configuring three burnout risk groups which could be considered as antecedents of burnout when defined in classical terms [38]. Therefore, the total score of each MBI dimension (e.g., exhaustion, cynicism, and efficacy) was considered as a dependent variable (DV), whereas the general factor scores of the BCSQ-36—or the subscale factor scores of the BCSQ-12—were included as independent variables (IVs). Multiple determination coefficients (R2) were calculated to evaluate the explanatory power of the models. The individual contribution of the IVs was estimated by standardized slope coefficients (Beta). The Wald test was used to evaluate the significance of the contribution of each IV.

Participants situated above the 75th percentile for each burnout subtype/subscale were considered to have “high scores”, whereas those below the 75th percentile were considered to have “low scores”, as it has been already done before [55], to have comparable results with previous research. Using simple binary logistic regression models to yield odds ratios (ORs) with a 95% confidence interval (95% CI), we conducted a bivariate analysis to assess the potential association between the subtypes and the sociodemographic and occupational variables referred to above. The level of significance of the associations was evaluated using the Wald test. Those variables that showed a value of *p* < 0.1 as a result of the exploratory bivariate analysis were later included in a multivariate model to estimate adjusted ORs and their 95% CIs.

The statistical analysis was conducted using SPSS v22.0 and Mplus v8.1 packages. All the tests were two-sided and were performed with a significance level of α < 0.05.

## 3. Results

A total of 407 participants answered the survey with no missing data and were included in the study. Their sociodemographic and occupational characteristics can be found in Table 1. Most were females (84.5%), with a mean age of 41.09 years (SD = 10.09); had a partner (70.3%); had children (60.7%); and were professionals with a salary (61.2%). Community health workers accounted for 57.2% of the sample, while 25.1% were nurses and 17.7% were physicians. The mean length of service in the PC setting was 17.19 years (SD = 9.81), with a mean of 5.47 years (SD = 5.53) at the current job. Most participants had full-time and permanent contracts (89.9% and 90.4%, respectively). Nonetheless, 84.0% of them reported having economic difficulties at some point. They worked for a mean of 39.25 h per week (SD = 26.81). Those who have had sick leave absences during the last year (35.6%) showed a mean of 19.0 sick leave days (SD = 51.83).

### 3.1. Psychometric Properties of the Brazilian BCSQ

#### 3.1.1. Long BCSQ-36

The one-factor solution showed the worst fit in all the burnout subtype scales. The three-correlated factors models had acceptable fit indices (Table 2), with adequate loadings and uniqueness (Table 3), omega for the total scales/subscales (Table 4), AVE values and inter-factor correlations: frenetic ((AVE: ambition = 0.59, overload = 0.58, involvement = 0.54), (φ: ambition‒overload = 0.60, *p* < 0.001; ambition‒involvement = 0.68, *p* < 0.001; overload‒involvement = 0.59, *p* < 0.001)); under-challenged ((AVE: indifference = 0.51, lack of development = 0.58, boredom = 0.59), (φ: indifference‒lack of development = 0.73, *p* < 0.001; lack of development‒boredom = 0.88, *p* < 0.001; indifference‒boredom = 0.81, *p* < 0.001)); worn-out ((AVE: lack of acknowledgement = 0.55, neglect = 0.66, lack of control = 0.52), (φ: lack of acknowledgement‒neglect = 0.41, *p* < 0.001; lack of acknowledgement‒lack of control = 0.67, *p* < 0.001; neglect‒lack of control = 0.46, *p* < 0.001)). Nevertheless, the bifactor model presented the best fit to the data (Table 2), explaining 78.4% of the variance in the frenetic scale, 78.5% in the under-challenged, and 79.3% in the worn-out, showing adequate loadings in their corresponding general factor (Table 3 and Table 4).

The mean inter-item Pearson’s correlation for the frenetic scale was 0.42, with a mean of item-rest coefficients of 0.60 (ambition = 0.69, overload = 0.68, and involvement = 0.65); for the under-challenged scale was 0.49, with a mean of item-rest coefficients of 0.67 (indifference = 0.59, lack of development = 0.69, boredom = 0.72); and for the worn-out scale was 0.37, with a mean of item-rest coefficients of 0.56 (lack of acknowledgement = 0.61, neglect = 0.75, lack of control = 0.63). As can be seen in Table 4, the omega hierarchical, construct replicability, and factor determinacy for the common general factors were adequate in all the burnout subtypes. However, they were rather scarce for the subscales once the effect of the corresponding general factor was partialized. The ECV values were as follows: frenetic = 0.62, under-challenged = 0.76, and worn-out = 0.53. The PUC reached a value of 0.73 in all the burnout subtypes scales.

#### 3.1.2. Short BCSQ-12

The one-factor solution of the short BCSQ-12 showed the worst fit indices, while the three-correlated factors model presented an appropriate fit (Table 2), as well as factor loadings, uniqueness terms, and AVE values (Table 5), explaining 70.6% of the total variance. Inter-factor correlations (φ) were as follows: overload‒lack of development = 0.12, *p* = 0.083; overload‒neglect = 0.10, *p* = 0.098; and lack of development‒neglect = 0.42, *p* < 0.001. The brief BCSQ-12 did not support the bifactor model structure (this solution did not converge). The mean inter-item Pearson’s correlations were overload = 0.57, lack of development = 0.58, and neglect = 0.66. The mean of item-rest coefficients were overload = 0.68, lack of development = 0.69, and neglect = 0.75. As can be seen in Table 5, the omega values for the subscales as well as their corresponding construct replicability and factor determinacy indices were appropriate.

### 3.2. Long/Short Brazilian BCSQ and other Constructs

The intercorrelations among the factorial scores of the Brazilian version of the long BCSQ-36 were frenetic‒under-challenged (r = 0.02, *p* = 0.694), frenetic‒worn-out (r = 0.12, *p* = 0.017), and under-challenged‒worn-out (r = 0.56, *p* < 0.001). The correlations between the long‒short BCSQ versions were frenetic‒overload (r = 0.71, *p* < 0.001; adj-r = 0.59, *p* < 0.001), under-challenged‒lack of development (r = 0.87, *p* < 0.001; adj-r = 0.72, *p* < 0.001), and worn-out‒neglect (r = 0.60, *p* < 0.001; adj-r = 0.52, *p* < 0.001). Table 6 shows the correlations between the factorial scores of the general common factor of the long BCSQ-36 as well as those of the BCSQ-12 subscales and the psychological health-related variables included in the present study.

Table 7 shows the explanatory power of the general factors of the BCSQ-12 and BCSQ-36 in relation to the MBI-GS. As a first step, we included the sociodemographic and occupational variables of sex, age, hours worked per week, and job position as a way to control for possible unbalanced characteristics. This first step only explained 5% (*p* < 0.001) of exhaustion, 2% (*p* = 0.149) of cynicism, and 9% (*p* < 0.001) of efficacy. The inclusion of the BCSQ-12 factorial scores improved the explanatory power to 29% (*p* < 0.001) of exhaustion, 34% (*p* < 0.001) of cynicism, and 17% (*p* < 0.001) of efficacy, while inclusion of the BCSQ-36 latent general common factors achieved values of 35% (*p* < 0.001) for exhaustion, 39% (*p* < 0.001) for cynicism, and 23% (*p* < 0.001) for efficacy. The standardized slopes of the burnout subtypes to explain the MBI-GS dimensions can be seen in Table 7. The slopes were consistent in terms of significance across both the long and short versions of the BCSQ, with the only exception of neglect/worn-out when explaining cynicism—neglect significantly explained cynicism, but the worn-out general common factor did not significantly explain that facet of the syndrome. As shown in Table 7, the different burnout subtypes presented distinct patterns of relationship with regard to the classical burnout dimensions.

Participants with high scores in the frenetic subtype were more likely to be professionals with a salary (OR = 1.68; 95% CI = 1.03–2.76; *p* = 0.040) and to work more than 40 h per week (OR = 2.31; 95% CI = 1.02–5.21; *p* = 0.044) when compared with participants with low scores in the frenetic subtype. Participants with high scores in overload were more likely to work more than 40 h per week (OR = 2.69; 95% CI = 1.20–6.06; *p* = 0.017) when compared with participants with low scores in overload.

Participants with high scores in the under-challenged subtype were more likely to be professionals with a salary (OR = 1.70; 95% CI = 1.04–2.77; *p* = 0.034) when compared with participants with low scores in the under-challenged subtype. Participants with high scores in lack of development were more likely to be more than 50 years old (OR = 2.10; 95% CI = 1.08–4.09; *p* = 0.030) and were also more likely to be males (OR = 2.20; 95% CI = 1.14–4.25; *p* = 0.019) when compared with participants with low scores in lack of development.

Participants with high scores in the worn-out subtype were more likely to be more than 50 years old (OR = 4.58; 95% CI = 1.89–11.10; *p* = 0.001), were less likely to be a nurse (vs. a physician, OR = 0.35; 95% CI = 0.15–0.79; *p* = 0.011), were more likely to have more years of service (6‒16 years (OR = 6.11; 95% CI = 2.37–15.78; *p* < 0.001) and >16 years (OR = 7.07; 95% CI = 2.32–21.59; *p* = 0.001)), and were more likely to have used sick leave days during the past year (OR = 0.39; 95% CI = 0.24–0.64; *p* < 0.001) compared with those with low scores in the worn-out subtype. Participants with high scores in neglect were more likely to not have a relationship (OR = 1.71; 95% CI = 1.05–2.78; *p* = 0.032) and were more likely to have economic difficulties (almost always (OR = 2.53; 95% CI = 1.12–5.72; *p* = 0.025) and always (OR = 2.70; 95% CI = 1.16–6.29; *p* = 0.021)) when compared with those with low scores in neglect (see Appendix A).

## 4. Discussion

### 4.1. Psychometrics of the Brazilian BCSQ

The main aim of the present work was to validate a Brazilian version of the BCSQ. So far, it has not been possible to measure the different burnout subtypes using the Portuguese language in general and in Brazilian contexts in particular. This was a need of clinicians and those in charge of the implementation of programmes because not all subjects suffering from the burnout syndrome present the same risk factors, characterization, and prognosis [20,21,25,28,31,32,39]. In our study, we have seen that the Brazilian versions of the BCSQ, both in its long (BCSQ-36) and short (BCSQ-12) formats, were suitable for use with Brazilian PC professionals.

The structure of the long BCSQ-36 behaved consistently throughout CFA. All the items loaded perfectly on their corresponding factors following the three-factor original design for each burnout subtype, and they were well explained with acceptable fit indices. The scale and subscale composite reliabilities were appropriate in all the cases, and following the Fornell and Larcker criteria [72], it was possible to establish convergent validity in the construct level for the frenetic, under-challenged, and worn-out subscales, although discriminant validity was only observed in the frenetic and worn-out subscales. This suggested that the under-challenged subscales might be conceptually very close. In fact, the under-challenged was the burnout scale with better adjustment to the bifactor model, highlighting the presence of a strong common factor.

All the burnout subtype scales of the long BCSQ-36 presented the best fit to the bifactor model, which allowed us to assess the features of the common general factor and relative strength of general to group factors. The general factor loadings were appropriate, with positive and significant values in all cases, explaining a high percentage of the total variance and of the reliable variance [69]. Likewise, construct replicability and factor determinacy were good in all the general factors, which means that they were well defined, and score estimates unambiguously reflected the latent levels they attempted to estimate [69,70,74]. On the contrary, the percent of reliable variance, factor determinacy, and construct replicability in subscales out of the influence of the common factor was low, suggesting the presence of robust common general factors but less powerful subscales in all the burnout subtypes. About ⅔ of explained common variance was attributable to the general frenetic factor, with ¾ belonging to the general under-challenged factor and with ½ belonging to the general worn-out factor. In the context of uncontaminated correlations and the other indices, we can conclude that the bifactor model could be accepted as an appropriate representation of all the burnout subtype scales and that the burnout subtypes could be treated in a unidimensional way in Brazilian PC professionals. Nevertheless, only the under-challenged subtype might be strictly considered as essentially unidimensional [69]. This is because the frenetic and worn-out subscales seemed to add some particularities beyond their corresponding general factor, although none of the subscales explained a sufficient portion of reliable variance beyond the general factor.

The structure of the short BCSQ-12 was consistent with the three-correlated factors model throughout CFA, showing appropriate fit indices. Items were loaded onto their corresponding factor with high values, explaining a high percentage of the total variance. The pattern of correlations between the subscales showed that overload was conceptually far away from lack of development and neglect, which were conceptually closer to each other—this result was also observed when analysing the relationships among the BCSQ-36 general factors. Previous studies including Spanish university workers [32] and healthcare students [56] have obtained this same result, suggesting that the under-challenged and worn-out subtypes might present some aspects in common as opposed to the frenetic subtype, such as a lower interest in tasks [20,21].

The bifactor structure was not supported by the BCSQ-12, and thus, it was not possible to evaluate the characteristics of a possible general factor. This is in line with the background of the BCSQ-12 [32,56] because this short version presents a scale composed by three factors (e.g., overload, lack of development, and neglect); each one of those represent a distinct burnout subtype with great discriminative force [31], and thus, a unidimensional score does not make theoretical sense. In these conditions, the three-correlated factors model was considered the best representation of this short version, which showed appropriate average variance extracted and convergent-discriminant validity [72]. In addition, construct replicability, factor determinacy, and composite reliability were good in all the components [70,74], reinforcing the plausibility of the three-correlated factors model for the BCSQ-12 in Brazilian PC professionals.

### 4.2. The BCSQ and other Constructs

The long BCSQ-36 general factors were highly convergent with the corresponding BCSQ-12 subscales, which means that the short version summarized fairly well the general factors of the long version. In general and according to the regression analyses developed, a considerable and significant conceptual overlap between the MBI scores and burnout subtypes was observed. However, the factors composing the short BCSQ-12 version replicated more clearly than the BCSQ-36 subtypes the results of convergence with the MBI classical definition found in previous research [31,32,56] because overload was mainly related to exhaustion and lack of development to cynicism, neglect being the factor more associated with (lack of) efficacy.

When evaluating the raw relationships between the long BCSQ-36 general common factors and the health-related constructs (e.g., anxiety, depression, affectivity, and guilt), the correlations observed reflected certain deterioration in terms of symptoms from lowest to highest degree from the frenetic going through the under-challenged to the worn-out in parallel with their expected or theoretical disengagement at work. However, this pattern was not observed in the same way according to the classical burnout and work engagement dimensions because both the under-challenged and worn-out subtypes appeared to be rather similarly affected by the classical burnout symptoms and lack of engagement.

It is striking that efficacy was significantly and positively related to the frenetic subtype. There is a strong debate that challenges the view of (lack of) efficacy, measured with a reversed efficacy scale, as a constituent of burnout [75]. In fact, exhaustion and cynicism have been considered the true core dimensions of burnout, thereby excluding (lack of) efficacy [76]. However, efficacy helps us to characterize the frenetic subtype, which appears as an active and committed profile [39] that at the same time suffers from exhaustion, maybe by moving into the sensitive area where engagement and workaholism converge [31,77]. This could also be why it presents significant relationships with positive and negative affects at the same time, as we have observed in the present study, in line with a previous work with Spanish PC professionals [40].

We have observed that cynicism was related to both the under-challenged and worn-out profiles, but it reached special strength in the former that also showed a significant exhaustion, (lack of) efficacy, absence of positive affect, and presence of negative affect. The under-challenged subtype seems to be a very affected profile that could be using cynicism as a dysfunctional coping strategy based on cognitive avoidance [39], leading to depersonalized attitudes, as it has been observed in physicians and nurses in other works [78,79]. The distance from obligations and detachment from tasks, workmates, and patients this attitude entails might reduce the adaptive ability at work, eroding possible positive affectivity derived from a job well done and from positive relationships, promoting negative affectivity, and exacerbating a lack of development, as has been previously proposed in PC physicians [80].

The worn-out subtype, in addition to cynicism, also presented significant levels of exhaustion and (lack of) efficacy as well as the absence of positive affect and presence of negative affect. The worn-out subtype appears to be also a very affected profile in terms of symptoms, as observed in the raw analyses referred to above and obtained in previous research [21,25,31,32,39]. This subtype has demonstrated a passive coping tendency [39] that has usually been observed when high levels of job strain are present [81]. It has been said that this trend might result from inconsequential histories of contingencies regarding control and awards in the workplace [28], all of which may be causing serious difficulties in alleviating stress [82].

The development of the burnout condition has been explained longitudinally by the degree of dedication at work, progressing from more to less (from enthusiasm to apathy), in an attempt to adopt a certain distance to alleviate the excess of activity but at the cost of entering into the spiral of erosion of illusion and a pattern of perceived stress due to frustration [21,83,84]. If we apply this theoretical framework—which understands burnout as a progressive impairment in levels of commitment—to the subtype conceptualization, one might interpret the burnout profiles as different stages in the development of the syndrome [21,25,31]. Nevertheless, as we have seen, our results only partially fit with this assumption because, although this gradation seemed to be observed in the degree of affection in the psychological health-related variables and the frenetic was the more engaged and less affected subtype in terms of burnout, the other two profiles were rather similarly affected by burnout symptoms and even the under-challenged might suffer from greater lack of engagement.

Following the dynamics of protective inhibition of self-regulation and motivation (PRISM) framework [85], employees who keep responding to challenges with high engagement (e.g., frenetic subtype) may reach a phase of exhaustion over time. Engagement involves costly physiological mobilization of resources through sympathetic activation [86], and thus, if engagement coping continues during initial stages of exhaustion, PRISM may set in to limit the health consequences of chronic hyper-engagement. PRISM decreases the level of perceived resource availability, leading to lowering of the maximum amount of resources one wants to invest in a goal and to perceiving that success on the goal requires a larger amount of resources relative to the perceived level of available resources. Together, this mechanism increases the likelihood of abandonment (e.g., worn-out subtype) to prevent negative health consequences of continued hyper-engagement. However, the under-challenged subtype situation appears to trigger a different—and importantly, according to our results, not less serious—process that may lead to burnout. Being under-challenged means that intrinsic and potential motivation will be low. Consequently, although the under-challenged individual will typically need to exert only little effort, when occasionally higher levels of effort are required, the level of potential motivation is quickly reached and PRISM sets in [85]. If this mechanism happens often, PRISM will further decrease the level of potential motivation, leading to cynicism. Thus, the typological perspective on burnout could be explained along different deterioration processes and psychological alterations which burnout could involve in its developmental course, in general, by reducing levels of motivation [87]. The traditional MBI model appears to not discriminate different burnout processes, and especially, it overlooks those risk factors derived from the under-challenged profile [25,38]. By contrast, combining PRISM with the typological approach could drive future research to explain different dynamics in developing the syndrome.

We have also seen that all the subtypes were significantly related to anxiety and depression. In recent years, there has been disagreement on whether burnout and anxiety/depression are the same or different constructs; as it seems, there is some overlap between them because they share some common characteristics, such as impaired concentration and loss of interest [88,89]. In line with a recent study that considered burnout from the MBI classical perspective [90], our results suggests the burnout subtypes and anxiety/depression seem to be different but related constructs, although more research is needed to analyse the nature of specific symptoms in which they are overlapped. Interestingly, the presence of guilt at work was made to manifest with significant coefficients in all the burnout subtype scales. It has been said that this variable could play a major role in the development and chronification of the syndrome by means of a positive feedback mechanism [64]. Thus, clarifying the specific role feelings of guilt may play in the burnout development in Brazilian PC professionals could bring greater understanding of the clinical evolution of the syndrome in this population, adding new psychotherapeutic elements for consideration when designing clinical interventions [21].

### 4.3. Sociodemographic Related to the Subtypes

The frenetic subtype presented high values in Brazilian PC professionals with a salary (vs. volunteers) who worked more than 40 h per week. Previous research [55] has stated that the type of work contract might influence the use of more or less active coping strategies, which in turn could determine the level of commitment. As can be understood, PC volunteers could be investing lower levels of dedication than professionals and thus presenting lower levels in the frenetic profile. In addition, the number of hours worked per week has been proposed as a key factor in the configuration of the frenetic subtype and, above a certain limit, it might be contributing to the development of the syndrome by increasing exhaustion [55].

The under-challenged subtype showed high values in Brazilian PC professionals with a salary, and the presence of lack of development was more present in professionals with more than 50 years of age. In general, under-challenged workers are disenchanted and feel trapped in an occupational activity that produces no gratifications and lack of personal growth [55]. Therefore, one might suppose that those older Brazilian PC providers that could be highly dependent on their salary and thus were obliged to carry on certain routine duties that maybe did not satisfy them and, at the same time, were not able to follow possible desires to change their job, could be suffering from the under-challenged subtype [91].

Higher scores in the worn-out subtype were more often in professionals with more than 50 years of age, having more years of service, being physicians (vs. being nurses), and using sick leave days in the past year, and the presence of neglect was more present in workers that did not have a relationship and had economic difficulties. Some studies have suggested the risk of burnout is higher around the middle of the physician career and then decreases as their experience and abilities to cope with stress increases [92]. It has also been pointed that middle-aged general practitioners had a high risk of burnout compared with other age groups [93]. Other studies have shown that the greater seniority, the more the worn-out level presented [28], and this has been explained as a result of the negative impacts of inconsequential organizations and job environments throughout time [94]. In this sense, physicians seemed to be in a worse position compared with nurses, maybe due to a greater level of uncertainty that was not entirely or properly acknowledged. Supposing that the worn-out subtype might be a final state of the burnout syndrome, with a marked physical and mental deterioration [21,25,31,55], sickness absences are likely to occur in the presence of high worn-out scores due to the great amount of symptoms this kind of worker presents [94]. Interestingly, we have seen that having a stable relationship could be a protective factor of the worn-out burnout subtype, as it has also been found in university workers, probably due to the positive effect of social support [55]. Finally, the fact that subjects with high worn-out scores are more prone to having economic difficulties might be illustrating that, in spite of their possible seniority, they have not been able to reach a secure position in economic terms and thus have been subjected to certain helplessness.

### 4.4. Limitations

We acknowledge several limitations to our study that should be taken into consideration. First, we used self-reported questionnaires, which involves some potential risks, such as recall bias. Second, although we tried to make the invitation to participate in the study as extensive as possible, we are unable to determine how many professionals saw and read the invitation so selection bias cannot be completely ruled out (e.g., those subjects with the highest scores on the variables of interest, due to their own idiosyncrasy, could be absent to a greater extent). Third, the sample was not well balanced for sociodemographic and occupational characteristics, which posed a limitation when trying to extend the scope of results, although we controlled for them when developing multivariate regression models. In addition, our study was strengthened by the inclusion of physicians as well as other health PC personnel, which was very helpful for the characterization of the different burnout profiles in this population as a whole. Fourth, we evaluated the convergence between the BCSQ and the MBI-GS as a classical measure of burnout, but this was originally developed to be used in general occupational groups. Thus, an MBI survey that is more orientated towards human services (e.g., MBI-HSS) might have allowed a more precise assessment for our study group. Nevertheless, the MBI-GS has also been used in PC health services [40], and it was used in this study to have an estimation comparable with previous research using the BCSQ. Finally, this is a cross-sectional study, so we can neither test predictive validity and test–retest reliability nor be certain about causation of correlated factors.

## 5. Conclusions

This work provides evidence about acceptable psychometric properties for the BCSQ among Brazilian PC professionals (hypothesis 1). The characteristics of the general common factor of each burnout subtype suggests the plausibility of the use of a total score in each of the BCSQ-36 scales—although not in the case of the short BCSQ-12 version. We have observed that the burnout subtypes significantly explained the symptoms of burnout according to the classical definition in Brazilian PC professionals (hypothesis 2), although the level of relationships observed suggests they do not exactly reflect the same constructs (e.g., the burnout subtypes could rather constitute groups of risk factors of burnout as it is understood in classical terms) [38]. We have seen that the frenetic subtype was related to exhaustion through overload, the under-challenged to cynicism was related through lack of development, and the worn-out to (lack of) efficacy was related through neglect (hypothesis 3). This pattern of relationships was clearer in the BCSQ-12 than in the BCSQ-36. Thus, the BCSQ-12 remains as a brief proposal with a relative convergence regarding the classical burnout definition and with a high ability to differentiate the subtypes. We have seen that the subtypes were differentially associated with the psychological health-related variables, and these relationships marked a progressive deterioration from the frenetic to the under-challenged and worn-out, in parallel to their theoretical disengagement at work (hypothesis 4). It was observed that the frenetic was an engaged profile, and thus, it was opposed to both the under-challenged and worn-out, which shared a certain lack of engagement. This suggests the frenetic could be a prodrome that may end up being worn-out [95] while the under-challenged might follow a different—but not less relevant—way in the burnout development process. Finally, we have found the burnout subtypes were differentially related to sociodemographic and occupational variables (hypothesis 5). These relationships suggest that intervene burnout subtypes are not only an individual clinical issue but also an organizational theme. For instance, some organizational aspects that should be guarded are the number of working hours (frenetic), the possibility of offering promotions (under-challenged), and job consolidation opportunities (worn-out).

Future research should focus on investigating how to act in the initial stages of burnout subtypes to prevent Brazilian PC professionals from suffering personal and social deterioration in the workplace and in what way interventions could include the organizational aspects of this specific context [96]. Some of the clinical implications a programme targeting the burnout subtypes should incorporate have been proposed theoretically in a previous work [21]. All in all, the importance of the availability of a Brazilian questionnaire to identify the burnout syndrome in its different clinical manifestations is clear to the PC settings, given the prevalence of this condition among PC personnel since, unfortunately, this syndrome seems to be quite widespread. We hope the present study serves as a bridge to facilitate implementation initiatives in the context of Brazilian PC services.

## Figures and Tables

**Table 1 ijerph-17-01081-t001:** Sociodemographic and occupational characteristics of study participants.

Variables	n (%)
Age, mean in years (SD)	41.09 (10.09)
<35	119 (29.2)
35–50	191 (46.9)
>50	97 (23.8)
Sex, female	344 (84.5)
Relationship, partnership/married	286 (70.3)
Number of children, none	160 (39.3)
Category	
Volunteer	158 (38.8)
Professional with a salary	249 (61.2)
Job position	
Physician	72 (17.7)
Nurse	102 (25.1)
CHW	233 (57.2)
Hours worked per week, mean in hours (SD)	39.25 (26.81)
<40	72 (17.7)
40	268 (65.8)
>40	64 (15.7)
Length of service, mean in years (SD)	17.19 (9.81)
<6	56 (13.8)
6–16	153 (37.6)
>16	197 (48.4)
Years at the same job, mean (SD)	5.47 (5.53)
<6	265 (65.1)
6–16	123 (30.2)
>16	18 (4.4)
Contract period	
Temporary	39 (9.6)
Permanent	368 (90.4)
Contract type	
Full-time	366 (89.9)
Part-time	41 (10.1)
Economic difficulties	
Never	65 (16.0)
Sometimes	153 (37.6)
Almost always	112 (27.5)
Always	77 (18.9)
Sick leave in the past year, yes	145 (35.6)
Sick leave days ^‡^	19.00 (51.83)

Note: Strata presented according to subsequent analyses. CHW: community health workers. ^‡^ Considering the group with sick leave days in the past year.

**Table 2 ijerph-17-01081-t002:** Fix indices for the frenetic, under-challenged, and worn-out BCSQ-36 and BCSQ-12 scales.

Scales/models	χ^2^	df	χ^2^/df	CFI	TLI	RMSEA	SRMR	AIC
**Frenetic**								
One factor (ref)	520.07	54	9.63	0.722	0.660	0.146 (0.134‒0.157)	0.094	15,581.45
Three correlated	168.00	51	3.29	0.930	0.910	0.075 (0.063‒0.088)	0.049	15,085.41
Bifactor model	127.87	42	3.04	0.949	0.920	0.071 (0.057‒0.085)	0.035	15,043.73
**Under-challenged**								
One factor (ref)	265.28	54	4.91	0.865	0.835	0.098 (0.086‒0.110)	0.061	16,362.76
Three correlated	186.93	51	3.67	0.913	0.887	0.081 (0.069‒0.094)	0.061	16,232.99
Bifactor model	97.21	42	2.31	0.965	0.944	0.057 (0.042‒0.072)	0.033	16,078.12
**Worn-out**								
One factor (ref)	736.77	54	13.64	0.582	0.489	0.176 (0.165‒0.188)	0.121	16,744.43
Three correlated	122.27	51	2.40	0.956	0.943	0.059 (0.045‒0.072)	0.053	15,937.22
Bifactor model	87.93	42	2.09	0.972	0.956	0.052 (0.037‒0.067)	0.042	15,907.40
**BCSQ-12 ***								
One factor (ref)	1280.25	54	23.71	0.215	0.041	0.236 (0.225‒0.248)	0.243	17,493.23
Three correlated	85.97	51	1.69	0.978	0.971	0.041 (0.025‒0.056)	0.035	15,923.81

Note: One factor (ref): one first-order factor solution taken as reference. Three-correlated: three correlated first-order factors model. Bifactor model: three orthogonal factors and a general factor integrating their commonalities. χ2: chi-squared. df: degrees of freedom. CFI: comparative fit index. TLI: Tucker-Leis index. RMSEA: root mean square error of approximation (90% CI). SRMR: standardized root mean square residual. AIC: Akaike information criterion. * The BCSQ-12 bifactor solution did not converge.

**Table 3 ijerph-17-01081-t003:** Descriptive and psychometric characteristics of the Brazilian BCSQ-36.

						3-Factors	Bifactor		
Scale/Subscale/Item	Mn	SD	skew	kurt	λ	δ		G	δ
**Frenetic**	4.66	0.97	−0.28	1.20					
Ambition	4.65	1.24	−0.32	0.22					
Item 1	4.54	1.55	−0.42	−0.19	0.67 *	0.55	0.55 *	0.48 *	0.47
Item 4	4.78	1.50	−0.52	−0.08	0.73 *	0.47	0.39 *	0.53 *	0.48
Item 7	4.76	1.41	−0.28	−0.22	0.83 *	0.31	0.27 ^†^	0.62 *	0.29
Item 10	4.54	1.50	−0.24	−0.40	0.83 *	0.31	0.57 *	0.61 *	0.26
Overload	4.13	1.32	−0.04	−0.43					
Item 2	4.63	1.56	−0.20	−0.56	0.66 *	0.57	0.37 *	0.61 *	0.58
Item 5	3.85	1.64	0.18	−0.61	0.81 *	0.35	0.50 *	0.66 *	0.37
Item 8	3.98	1.67	0.03	−0.69	0.80 *	0.37	0.65 *	0.80 *	0.32
Item 11	4.05	1.56	0.03	−0.58	0.77 *	0.41	0.61 *	0.51 *	0.38
Involvement	5.21	0.96	−0.83	3.16					
Item 3	5.37	1.20	−0.69	0.52	0.75 *	0.44	0.39 *	0.62 *	0.46
Item 6	5.34	1.20	−0.71	1.50	0.77 *	0.40	0.40 *	0.65 *	0.41
Item 9	5.13	1.22	−0.80	1.81	0.75 *	0.44	0.42 *	0.50 *	0.43
Item 12	4.99	1.11	−0.59	1.70	0.66 *	0.56	0.51 *	0.49 *	0.50
**Under-challenged**	3.35	1.20	0.14	−0.01					
Indifference	2.88	1.13	0.45	0.51					
Item 13	2.77	1.46	0.65	0.28	0.69 *	0.53	0.59 *	0.47 *	0.44
Item 16	2.59	1.39	0.84	0.82	0.83 *	0.32	0.54 *	0.59 *	0.30
Item 19	3.88	1.70	−0.02	−0.60	0.60 *	0.64	0.02	0.74 *	0.53
Item 22	2.28	1.27	0.99	1.39	0.71 *	0.50	0.46 *	0.63 *	0.49
L. Development	3.71	1.46	0.07	−0.44					
Item 14	3.72	1.77	0.18	−0.72	0.72 *	0.48	0.68	0.67 *	0.20
Item 17	3.42	1.61	0.36	−0.25	0.70 *	0.52	0.16 ^‡^	0.76 *	0.53
Item 20	4.12	1.89	−0.10	−0.97	0.81 *	0.34	0.42	0.68 *	0.31
Item 23	3.60	1.79	0.21	−0.82	0.82 *	0.34	0.10	0.72 *	0.35
Boredom	3.45	1.40	0.27	−0.18					
Item 15	3.65	1.73	0.20	−0.72	0.76 *	0.42	0.11	0.77 *	0.44
Item 18	3.46	1.71	0.29	−0.71	0.71 *	0.50	0.15	0.55 *	0.40
Item 21	3.27	1.63	0.43	−0.31	0.81 *	0.35	0.25	0.80 *	0.35
Item 24	3.42	1.71	0.31	−0.61	0.80 *	0.36	0.45 ^‡^	0.76 *	0.22
**Worn-out**	3.74	1.02	−0.11	0.76					
L. Acknowledgement	4.04	1.38	0.07	−0.29					
Item 25	3.68	1.73	0.25	−0.66	0.71 *	0.50	0.42 *	0.55 *	0.53
Item 28	3.82	1.81	0.19	−0.80	0.41 *	0.83	0.20 ^‡^	0.46 *	0.85
Item 31	4.39	1.74	−0.10	−0.81	0.79 *	0.37	0.46 *	0.62 *	0.42
Item 34	4.27	1.75	−0.08	−0.75	0.94 *	0.11	0.78 *	0.34 *	0.02
*Neglect*	2.64	1.14	0.64	1.24					
Item 26	2.76	1.39	0.57	0.13	0.79 *	0.37	0.64 *	0.41 *	0.38
Item 29	2.73	1.30	0.69	1.03	0.80 *	0.36	0.69 *	0.73 *	0.36
Item 32	2.30	1.16	0.72	1.24	0.80 *	0.36	0.70 *	0.60 *	0.35
Item 35	2.78	1.41	0.79	0.88	0.86 *	0.27	0.68 *	0.41 *	0.27
L. Control	4.53	1.29	−0.67	0.32					
Item 27	4.97	1.66	−0.72	0.04	0.73 *	0.47	0.51 *	0.73 *	0.36
Item 30	4.62	1.68	−0.54	−0.29	0.76 *	0.42	0.20	0.65 *	0.43
Item 33	3.60	1.56	0.13	−0.51	0.67 *	0.55	0.01	0.51 *	0.47
Item 36	4.95	1.59	−0.63	0.01	0.71 *	0.49	0.45 ^†^	0.61 *	0.43

Note: Mn: mean; SD: standard deviation. Skew: skewness. Kurt: kurtosis. λ: factorial loading. δ: uniqueness term. G: general factor. ^‡^
*p* < 0.05. ^†^
*p* < 0.01. * *p* < 0.001. P_75_ (75th percentile): Frenetic = 5.25; ambition = 5.50; overload = 5.00; involvement = 6.00; under-challenged = 4.08; indifference = 3.50; lack of development = 4.75; boredom = 4.25; worn-out = 4.33; lack of acknowledgement = 5.00; neglect = 3.25; lack of control = 5.25.

**Table 4 ijerph-17-01081-t004:** Reliability of the BCSQ-36 bifactor models.

Scale/*Subscale*	ω/ω_S_	ω_H_/ω_HS_	H	FDI	ECV
**Frenetic**	0.93	0.76	0.88	0.87	0.62
Ambition	0.81	0.32	0.54	0.56	
Overload	0.91	0.37	0.65	0.61	
Involvement	0.81	0.30	0.48	0.54	
**Under-challenged**	0.94	0.87	0.92	0.93	0.76
Indifference	0.84	0.26	0.55	0.51	
L. Development	0.88	0.17	0.53	0.41	
Boredom	0.85	0.09	0.26	0.29	
**Worn-out**	0.92	0.72	0.87	0.85	0.53
L. Acknowledgement	0.79	0.37	0.68	0.61	
Neglect	0.93	0.57	0.77	0.76	
L. Control	0.80	0.14	0.39	0.38	

Note: Percentage of Uncontaminated correlations (PUC) = 0.73. ω: McDonald’s omega for the total scale. ω_S_: omega subscale. ω_H_: omega hierarchical; ω_H_: omega hierarchical subscale. H: replicability index. FDI: factor determination index. ECV: explained common variance.

**Table 5 ijerph-17-01081-t005:** Confirmatory factor analysis (CFA) of the Brazilian short BCSQ-12.

Subscale/Item	ω_S_	H	FDI	AVE	λ	δ
**Overload**	0.85	0.86	0.93	0.58		
Item 2					0.64 *	0.60
Item 5					0.79 *	0.38
Item 8					0.82 *	0.33
Item 11					0.78 *	0.39
**L. Development**	0.85	0.88	0.94	0.59		
Item 14					0.78 *	0.39
Item 17					0.66 *	0.56
Item 20					0.85 *	0.28
Item 23					0.76 *	0.43
**Neglect**	0.89	0.89	0.95	0.66		
Item 26					0.79 *	0.37
Item 29					0.80 *	0.36
Item 32					0.81 *	0.35
Item 35					0.85 *	0.28

Note: Three-correlated factors solution. ωS: omega subscale. H: construct replicability index. FDI: factor determinacy index. AVE: average variance extracted; λ: factorial loading. δ: uniqueness. * *p* < 0.001.

**Table 6 ijerph-17-01081-t006:** Raw correlations between the BCSQ and other psychological health-related variables.

Psychological Health-Related Variables	Frenetic *(Overload)*	Under-Challenged *(L. Development)*	Worn-out *(Neglect)*
**Exhaustion**	0.19 * *(0.36 *)*	0.41 * *(0.34 *)*	0.51 * *(0.28 *)*
**Cynicism**	0.04 *(0.15* ^†^*)*	0.60 * *(0.52 *)*	0.42 * *(0.44 *)*
**Efficacy**	0.31 * *(0.11* ^‡^*)*	−0.21 * *(-0.14 ^†^)*	−0.20 * *(*−*0.26 *)*
**Vigour**	0.31 * *(0.09)*	−0.44 * *(−0.36 *)*	−0.40 * *(*−*0.39 *)*
**Dedication**	0.25 * *(0.08)*	−0.54 * *(−0.45 *)*	−0.37 * *(*−*0.33 *)*
**Absorption**	0.38 * *(0.24 *)*	−0.45 * *(−0.38 *)*	−0.28 * *(*−*0.32 *)*
**Anxiety**	0.14 ^†^ *(0.35 *)*	0.35 * *(0.29 *)*	0.42 * *(0.28 *)*
**Depression**	0.10 ^‡^ *(0.34 *)*	0.29 * *(0.24 *)*	0.35 * *(0.31 *)*
**Positive Affect**	0.31 * *(0.08)*	−0.41 * *(*−*0.31 *)*	−0.35 * *(*−*0.37 *)*
**Negative Affect**	0.26 * *(0.38 *)*	0.35 * *(0.31 *)*	0.49 * *(0.32 *)*
**Guilt at work**	0.12 ^‡^ *(0.22 *)*	0.30 * *(0.33 *)*	0.38 * *(0.31 *)*

Note: Values are Pearson’s correlation coefficients. BCSQ scale/subscale scores are factorial scores. * *p* < 0.001; ^†^
*p* < 0.01; ^‡^
*p* < 0.05. The short BCSQ-12 subscales are in brackets and in italic.

**Table 7 ijerph-17-01081-t007:** Explanatory power of the BCSQ on the classical burnout dimensions.

	DV/IVs		ΔR^2^	R^2^	Se	F	df	p^a^
	**Exhaustion**	*Step 1*	ref.	0.05	8.18	4.57	5/400	<0.001
		*Step 2*	0.24	0.29	7.12	20.25	8/397	<0.001
		*Step 3*	0.30	0.35	6.83	26.34	8/397	<0.001
				**b**	**Se**	**Beta**	**t**	**p^b^**
*Step 2:* BCSQ-12	Overload			2.64	0.40	0.29	6.68	<0.001
	L. Development			1.89	0.32	0.29	5.83	<0.001
	Neglect			0.90	0.39	0.11	2.32	0.021
*Step 3:* BCSQ-36	Frenetic			1.75	0.52	0.14	3.34	0.001
	Under-challenged			3.23	0.66	0.25	4.88	<.001
	Worn-out			3.28	0.50	0.33	6.51	<.001
	**DV/IVs**		**ΔR^2^**	**R^2^**	**Se**	**F**	**df**	**p^a^**
	**Cynicism**	*Step 1*	ref.	0.02	6.11	1.64	5/400	0.149
		*Step 2*	0.32	0.34	5.03	25.60	8/397	<0.001
		*Step 3*	0.37	0.39	4.84	31.52	8/397	<0.001
				**b**	**Se**	**Beta**	**t**	**p^b^**
*Step 2:* BCSQ-12	Overload			0.36	0.28	0.05	1.28	0.203
	L. Development			2.03	0.23	0.42	8.84	<0.001
	Neglect			1.34	0.27	0.23	4.91	<0.001
*Step 3:* BCSQ-36	Frenetic			0.28	0.37	0.03	0.75	0.452
	Under-challenged			5.36	0.47	0.57	11.41	<0.001
	Worn-out			0.59	0.36	0.08	1.65	0.100
	**DV/IVs**		**ΔR^2^**	**R^2^**	**Se**	**F**	**df**	**p^a^**
	**Efficacy**	*Step 1*	ref.	0.09	5.91	7.62	5/400	<0.001
		*Step 2*	0.08	0.17	5.67	9.79	8/397	<0.001
		*Step 3*	0.14	0.23	5.44	14.95	8/397	<0.001
				**b**	**Se**	**Beta**	**t**	**p^b^**
*Step 2:* BCSQ-12	Overload			0.85	0.32	0.13	2.70	0.007
	L. Development			−0.11	0.26	−0.02	−0.43	0.670
	Neglect			−1.50	0.31	−0.26	−4.88	<0.001
*Step 3:* BCSQ-36	Frenetic			3.00	0.42	0.32	7.19	<0.001
	Under-challenged			−0.86	0.53	−0.09	−1.62	0.106
	Worn-out			−1.25	0.40	−0.17	−3.12	0.002

DV: dependent variable; IV: independent variable; Step 1: includes age, sex, hours worked per week, and job position as IVs. Step 2 includes age; sex; hours worked per week; job position; as well as the standardised factorial scores of overload, lack of development and neglect as IVs. Step 3: includes age; sex; hours worked per week; job position; as well as the standardised latent general common factors of the frenetic, under-challenged, and worn-out subtypes as IVs. R^2^: determination coefficient. Se: standard error. F: Snedecor’s F. df: degrees of freedom. p^a^: *p*-value associated with the model adjustment. b: regression coefficient. Beta: standardised regression coefficient. t: Student’s *t* value related to the Wald test on regression coefficients. p^b^: *p*-value associated with the Wald test. ref.: category of reference.3.3. Sociodemographic Factors Related to the Burnout Subtypes

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
