# Peer review of "Frenetic, under-Challenged, and Worn-out Burnout Subtypes among Brazilian Primary Care Personnel: Validation of the Brazilian “Burnout Clinical Subtype Questionnaire” (BCSQ-36/BCSQ-12)"

_ijerph, 2020, doi:10.3390/ijerph17031081_

Round 1

Reviewer 1 Report

It is an excellent work. It merits to be published.

Only need to explain why the ethic permission has a date posterior (2016) to the realization of the screening (2015).

At Line 321-322 there is an Error! Reference source not found. Please resolve the error.

Author Response

-Only need to explain why the ethic permission has a date posterior (2016) to the realization of the screening (2015).

Thank you for this appreciation (by the way, the number of the code was wrong and we have corrected it). The date of the final approval (2016) is later than the date on which screening was conducted because the project included subsequent additions made after its initial approval. In the document, we reported the final date, but as can be seen in the attached pdf, the project included in the present study was already approved in 2015. We have included in the text that: “this date is later than the date on which screening was conducted because subsequent additions were made to the project after its initial approval; however, those modifications did not affect this work”.

-At Line 321-322 there is an Error! Reference source not found. Please resolve the error.

Thank you. This has been amended (It should have been “Table 1”).

Thank you so much for the time invested in reviewing the manuscript and for your appreciations.

Reviewer 2 Report

Title: mention the subtypes verbatim

Abstract: Given the novelty of the three new subtypes, the readers need some more background information in this regard; actually, while worn-out is somehow plausible, frenetic and under-challenged are not. Report briefly some characteristics of the original questionnaire. Report throughout the Abstract, if participants completed a Brazilian/Portuguese questionnaire.

“The pattern of relationships between the burnout subtypes and the psychological health-related outcomes suggested different burnout deterioration processes, and the socio-demographic data were differentially related to the subtypes;”; while this might be true, please be much more specific when reporting the pattern of results.

Introduction: The authors begin the Introduction with the description of the Brazilian Health Care system; while this is information is important to understand the background of the ‘plot’, I suggest to starting with the key topic of the present study, that is, with burnout and it facets, or the other way around: in my opinion, the paragraph beginning with “Burnout is one of the most important….” should be the beginning of the Introduction. Further, in my opinion, while it is not untrue that issues raised as regards the psychometric properties of different tools, the authors should emphasize the discussion, if and if so, to what extent burnout is considered a psychiatric disorder or cluster of its own, or if burnout is “simply” a specific dimension of major depressive disorders (Bianchi et al., 2015, 2016a, b, 2017a, b, c; Bianchi et al., 2017d; Bianchi et al., 2016c; Laurent et al., 2017; Schonfeld et al., 2016). In this view, please note that from 2020 on, the ICD-11 will have introduced burnout as a strictly work-load-related state of psychological exhaustion.

“….. a coping style focused on active problem-solving, with individuals employing a large number of working hours or getting involved in multiple tasks…”; perhaps the authors should mention that such kind of “active problem-solving” is highly dysfunctional and actually does not lead to ‘solve the problem’.

“under-challenged subtype”; the authors are in the need to explaining in much more details how lack of stimulation and motivation can match the key concept of burnout as a state of exhaustion as a result of continuous and strenuous work performance; or simply put: at a first glance, it does not seem that the concept under-challenged subtype does not match at all the key concept of burnout. This seems to be even more true, if we consider burnout as a specific case of major depressive disorders. In a similar vein, the authors should show some results/findings reporting correlation coefficients between these three novel variables and dimensions of anxiety and depression; otherwise, it is simply hard to believe that these new dimensions do reflect burnout traits, or rather other latent psychological constructs.

Methods; translation-backtranslation; it appears that the authors used the algorithms as proposed by Brislin (1986) and Beaton et al. (2000). Personally, I was impressed of the broad variety and the nice choices of the tools. Congrats on you! The statistical approaches were well-chosen. “All the tests were bilateral”; did you mean two-sided? Otherwise, the term ‘bilateral’ needs explanations.

Results: “… socio-demographic and occupational characteristics can be found in Error! Reference source not found.”; here, something went wrong.

Table 6; from correlation coefficients reported in this table, it appears that factors were highly inter-related and that most probably a one-factor or two-factors solution might be more appropriate. Results at pages 12-13, ll 403-412; try to report these results also in a table.

Discussion: well performed.

Conclusions: As often, it is a matter of taste, though, I suggest to trimming the Conclusion section as much as possible.

References

Beaton, D.E., Bombardier, C., Guillemin, F., Ferraz, M.B., 2000. Guidelines for the process of cross-cultural adaptation of self-report measures. Spine (Phila Pa 1976) 25(24), 3186-3191.

Bianchi, R., Schonfeld, I.S., Laurent, E., 2015. Burnout does not help predict depression among French school teachers. Scandinavian journal of work, environment & health 41(6), 565-568.

Bianchi, R., Schonfeld, I.S., Laurent, E., 2016a. The "Burnout" Construct: An Inhibitor of Public Health Action? Crit Care Med 44(12), e1252-e1253.

Bianchi, R., Schonfeld, I.S., Laurent, E., 2016b. The Dead End of Current Research on Burnout Prevalence. Journal of the American College of Surgeons 223(2), 424-425.

Bianchi, R., Schonfeld, I.S., Laurent, E., 2017a. Burnout or depression: both individual and social issue. Lancet (London, England) 390(10091), 230.

Bianchi, R., Schonfeld, I.S., Laurent, E., 2017b. Can we trust burnout research? Annals of oncology : official journal of the European Society for Medical Oncology 28(9), 2320-2321.

Bianchi, R., Schonfeld, I.S., Laurent, E., 2017c. On the overlap of vital exhaustion and depression. Eur Psychiatry 44, 161-163.

Bianchi, R., Schonfeld, I.S., Vandel, P., Laurent, E., 2017d. On the depressive nature of the "burnout syndrome": A clarification. Eur Psychiatry 41, 109-110.

Bianchi, R., Verkuilen, J., Brisson, R., Schonfeld, I.S., Laurent, E., 2016c. Burnout and depression: Label-related stigma, help-seeking, and syndrome overlap. Psychiatry research 245, 91-98.

Brislin, R., W., , 1986. The wording and translation of research instrument., in: Lonner, W.J., Berry, J.W. (Ed.) Field methods in cross-cultural research. SAGE, Beverly Hills, CA, pp. 137-164.

Laurent, E., Bianchi, R., Schonfeld, I.S., Vandel, P., 2017. Editorial: Depression, Burnout, and Other Mood Disorders: Interdisciplinary Approaches. Front Psychol 8, 282.

Schonfeld, I.S., Laurent, E., Vandel, P., Bianchi, R., 2016. Burnout and Depression in Psychiatric Residents. Can J Psychiatry 61(11), 737-738.

Author Response

-Title: mention the subtypes verbatim.

The title has been modified: “Frenetic, under-challenged and worn-out burnout subtypes among Brazilian primary care personnel: Validation of the Brazilian ‘Burnout Clinical Subtype Questionnaire’ (BCSQ-36/BCSQ-12)”  

-Abstract: Given the novelty of the three new subtypes, the readers need some more background information in this regard; actually, while worn-out is somehow plausible, frenetic and under-challenged are not. Report briefly some characteristics of the original questionnaire. Report throughout the Abstract, if participants completed a Brazilian/Portuguese questionnaire.

We have included in the abstract (considering the strong limitations in terms of number of words) that: “A new model of burnout has been developed to distinguish three subtypes: frenetic, under-challenged and worn-out, which are characterized as overwhelmed, under-stimulated and disengaged at work, respectively”, and also that: “Participants answered a Brazil-specific survey including the BCSQ-36/BCSQ-12,…”.

-“The pattern of relationships between the burnout subtypes and the psychological health-related outcomes suggested different burnout deterioration processes, and the socio-demographic data were differentially related to the subtypes;” while this might be true, please be much more specific when reporting the pattern of results.

We have clarified in the abstract (with the referred above limitations in the number of words) that: “The pattern of relationships between the burnout subtypes and the psychological outcomes suggested a progressive deterioration from the frenetic to the under-challenged and worn-out”.

-Introduction: The authors begin the Introduction with the description of the Brazilian Health Care system; while this is information is important to understand the background of the ‘plot’, I suggest to starting with the key topic of the present study, that is, with burnout and it facets, or the other way around: in my opinion, the paragraph beginning with “Burnout is one of the most important….” should be the beginning of the Introduction.

Many thanks for this suggestion. We fully agree and have consequently moved the paragraph to the beginning of the Introduction section, as requested by the reviewer, in order to provide a more logical introduction.

-Further, in my opinion, while it is not untrue that issues raised as regards the psychometric properties of different tools, the authors should emphasize the discussion, if and if so, to what extent burnout is considered a psychiatric disorder or cluster of its own, or if burnout is “simply” a specific dimension of major depressive disorders (Bianchi et al., 2015, 2016a, b, 2017a, b, c; Bianchi et al., 2017d; Bianchi et al., 2016c; Laurent et al., 2017; Schonfeld et al., 2016). In this view, please note that from 2020 on, the ICD-11 will have introduced burnout as a strictly work-load-related state of psychological exhaustion.

Many thanks for this important contribution. We have added to the introduction that: “In addition, there is a current trend that questions to what extent burnout is a psychiatric disorder or cluster on its own, proposing instead that burnout might “simply” be a specific dimension of major depressive disorders [12–16]”. We have also included in the introduction section that: “Other authors have highlighted that the burnout term is being widely used but poorly measured because the construct behind the classical definition could not be sufficiently valid due to a non-clinically based origin [17,18]. The latest revision of the International Classification of Diseases, ICD-11, introduces burnout as a work-related chronic state of stress and psychological exhaustion. However, it identifies this syndrome as an occupational phenomenon ‒ not a medical condition ‒ that is primarily related to the mismatch between the environment (i.e. demands) and the individual (i.e. resources that are necessary to develop the work meaningfully) [19]. One of the most significant downsides of the classical point of view of burnout, especially regarding the development of intervention strategies, is the fact that it evaluates all cases with a definition based on a scarce set of symptoms, whereas the psychosocial reality in which the syndrome develops tends to vary among cases [20,21]”.  

(the number of each reference is in the references section)

Bianchi, R., Verkuilen, J., Brisson, R., Schonfeld, I.S., Laurent, E., 2016. Burnout and depression: Label-related stigma, help-seeking, and syndrome overlap. Psychiatry research 245, 91-98.

Bianchi, R., Schonfeld, I.S., Laurent, E., 2017a. Burnout or depression: both individual and social issue. Lancet (London, England) 390(10091), 230.

Bianchi, R., Schonfeld, I.S., Laurent, E., 2017b. Can we trust burnout research? Annals of oncology: official journal of the European Society for Medical Oncology 28(9), 2320-2321.

Bianchi, R., Schonfeld, I.S., Laurent, E., 2017c. On the overlap of vital exhaustion and depression. Eur Psychiatry 44, 161-163.

Bianchi, R., Schonfeld, I.S., Vandel, P., Laurent, E., 2017d. On the depressive nature of the "burnout syndrome": A clarification. Eur Psychiatry 41, 109-110.

Eckleberry-Hunt J, Kirkpatrick H, Barbera T. The problems with burnout research. Acad Medicine. 2018;93(3): 367–370.

Farber B. Burnout in psychotherapist: Incidence, types and trends. Psychotherapy in Private Practice. 1990;8(1): 35–44.

Lancet. ICD-11. The Lancet, 2019, 393(10188):2275. doi: 10.1016/S0140-6736(19)31205-X.  

Messias E, Flynn V.  The tired, retired, and recovered physician: Professional burnout versus major depressive disorder. Am J Psychiatry. 2018;175(8): 716–719.

Montero-Marin J, Prado-Abril J, Demarzo M, García-Toro M, García-Campayo J. Burnout subtypes and their clinical implications: A theoretical proposal for specific therapeutic approaches. Revista de Psicopatología y Psicología Clínica. 2016;21(3): 231–242.

-“….. a coping style focused on active problem-solving, with individuals employing a large number of working hours or getting involved in multiple tasks…”; perhaps the authors should mention that such kind of “active problem-solving” is highly dysfunctional and actually does not lead to ‘solve the problem’.

We have added to the text that: “This subtype is associated with high levels of exhaustion and a coping style focused on active problem-solving ‒ which is highly dysfunctional and actually does not usually lead to ‘solving the problem’ when it is always used as the only coping strategy ‒ with individuals employing a large number of working hours or getting involved in multiple tasks [24–27]”.

-“under-challenged subtype”; the authors are in the need to explaining in much more details how lack of stimulation and motivation can match the key concept of burnout as a state of exhaustion as a result of continuous and strenuous work performance; or simply put: at a first glance, it does not seem that the concept under-challenged subtype does not match at all the key concept of burnout. This seems to be even more true, if we consider burnout as a specific case of major depressive disorders. In a similar vein, the authors should show some results/findings reporting correlation coefficients between these three novel variables and dimensions of anxiety and depression; otherwise, it is simply hard to believe that these new dimensions do reflect burnout traits, or rather other latent psychological constructs.

Many thanks for these important suggestions. We have added to the manuscript that: “At first glance, the under-challenged subtype does not seem to match the key concept of burnout, even more so if we consider burnout as a specific case of major depressive disorders. However, this profile seems to be also moderately related to exhaustion and lack of efficacy [25], creating a burnout risk group that might correspond to states of the syndrome that could have been overlooked in previous research [28]”. In the absence of other measures in previous studies, we have also included that: “… it is important to highlight the fact that the three burnout subtypes have presented adequate discriminative values regarding general negative affectivity states using the PANAS questionnaire in PC healthcare personnel, with Pearson’s correlation coefficient values ranging from r = .24 to .29 [29]”. 

-Methods; translation-back translation; it appears that the authors used the algorithms as proposed by Brislin (1986) and Beaton et al. (2000). Personally, I was impressed of the broad variety and the nice choices of the tools. Congrats on you! The statistical approaches were well-chosen. “All the tests were bilateral”; did you mean two-sided? Otherwise, the term ‘bilateral’ needs explanations.

Thank you for these comments. Yes, exactly, we used the translation-back translation algorithms referred to by the reviewer and we have now inserted the correct references in the text; and also we have changed “bilateral” to “two-sided”, which was the intended meaning.

-Results: “… socio-demographic and occupational characteristics can be found in Error! Reference source not found.”; here, something went wrong.

Sorry. It has been amended (it should have been “Table 1”).

-Table 6; from correlation coefficients reported in this table, it appears that factors were highly inter-related and that most probably a one-factor or two-factors solution might be more appropriate. Results at pages 12-13, ll 403-412; try to report these results also in a table.

Many thanks for this comment. As the reviewer suggests, correlation coefficients reported in Table 6 were relatively highly inter-related. Nevertheless, we have to take into account the fact that these coefficients were the result of raw correlations. We have reported the multiple regression results in a new table (Table 7), as requested by the reviewer, where we show the different role that each burnout subtype seems to play regarding the classical burnout dimensions, after including all of them at the same time in the models, and subsequent adjustment. Tolerance values of the burnout subtypes (BCSQ-12 and BCSQ-36) in these models were ≥.95, thus suggesting absence of collinearity. Therefore, according to the results of the fit of the structural models tested and also to the multiple regression analyses referred to here, we think the best way to represent the different burnout subtypes is using the bifactor model (BCSQ-36) and the three-correlated factors model (BCSQ-12). Both of them allow us to have one single factor for each subtype (in its long or short version), which have different features in terms of burnout.   

-Discussion: well performed.

Thank you so much.

-Conclusions: As often, it is a matter of taste, though, I suggest to trimming the Conclusion section as much as possible.

As suggested by the reviewer, we have removed several sentences from the Conclusion section to condense it as much as possible without leaving out important ideas.

Thank you so much for the time invested and the very relevant contributions suggested, which have helped us to improve the quality of the manuscript.

Reviewer 3 Report

The paper contains results of the psychometric properties of the long and short Brazilian versions of the Burnout Clinical Subtypes Questionnaire (BCSQ-36 and BCSQ-12). A cross-sectional design was employed through a convenience sample (n = 407) composed of health care practitioners and volunteers. The underpinning theory of the two versions of BCSQ was briefly described in the introduction section.

The English language is sufficiently appropriate and understandable.

I do agree with the authors idea to focus on importance of trusted tools assessing burnout. In my experience, the MBI has shown some weakness point to assess risk of burnout among health care practitioners. Therefore, I think that results of these paper could provide an advance in current knowledge.

However, there are some weakness in the paper that should be address by the authors. Hence, I include the following considerations about the manuscript in its current version:

- Page 2, lines 71-74: I am not sure to understand what the authors mean by the sentence “Several studies have identified chronic stress and burnout syndrome as relevant issues among PC personnel in Brazil and their relation to the characteristics of the working environment”. Is that primary healthcare practitioners have more risk to burn out? A more detailed description of the results of these studies could help the readers to understand the theoretical and empirical framework.

- pages 2-3, lines 51-145: I feel that the logic of the progression is not clear to follow for the readers. A more precise delineation of the most important studies conducted in Brazil is needed to understand why the authors have conducted this research.

- page 2, lines 79-85 and page 5, lines 214-223: the authors refer to the MBI-General Survey, that is a measure of burnout in occupational groups outside human services work (Schaufeli, Leiter, Maslach, & Jackson, 1996). Why they did not employ the MBI? Is not available in Brazil? I feel that a deeper rationale is needed to realize this choice.

- page 5, lines 232-238: the HADS is generally used in clinical settings with hospitalized patients. Why did the authors make this choice?

- page 7, lines 299-307: I have some doubts about these multiple linear regressions. As stated by authors, the BCSQ is referred to a new model of burnout overlapping the Maslach’s theory. How can BCSQ factors and subscales be treated as independent variables in multiple linear regression analysis? Theoretical rationale for this choice should be explained by authors.

- page 7, lines 321-322: could authors explain what it is meant by “Error” Reference source not found..”?

-pages 7-8, lines 322-333: the sample is not well balanced for socio-demographic and occupational characteristics. This limit should be point out in the limitations paragraph (page 17, lines 618-626). Moreover, it could be useful to evaluate the role of these variables on the results by insert them in the multiple regression analysis as control variables in the first step.

- There are not results about retest of the two versions of BCSQ. Why did not include them in the study two on a part of the sample?

My recommendation for the Editor is to accept after revision.

Author Response

- Page 2, lines 71-74: I am not sure to understand what the authors mean by the sentence “Several studies have identified chronic stress and burnout syndrome as relevant issues among PC personnel in Brazil and their relation to the characteristics of the working environment”. Is that primary healthcare practitioners have more risk to burn out? A more detailed description of the results of these studies could help the readers to understand the theoretical and empirical framework.

We have merged this response with the following one because they are closely related

- pages 2-3, lines 51-145: I feel that the logic of the progression is not clear to follow for the readers. A more precise delineation of the most important studies conducted in Brazil is needed to understand why the authors have conducted this research.

We agree that the logic of the progression in the introduction section was not entirely clear. We have now restructured the introduction section, moving the paragraphs to present a more linear rationale. For this, we now begin by presenting the burnout construct, its shortcomings and its needs of improvement and refinement, and also the alternative that involves the introduction of the burnout subtypes in the field; we then explain the specific situation of the PC healthcare context in Brazil, to understand the need for and importance of having and introducing new tools to measure and study burnout syndrome in this population. In this regard, we have added to the manuscript that:

 “The increase in PC coverage since the late nineties has been remarkable in Brazil, from zero in 1993 to about 64% of Brazilian population in 2019 (representing over 134 million people, mostly low income or underserved), and marked a decrease in inequities of access that was translated into greater user satisfaction and a reduction in infant and adult mortality, in addition to adult complications related to chronic non-communicable diseases. However, in order to achieve universality, equity, and sustainability, the Family Health Strategy still faces many challenges that include chronic underfunding, heterogeneity of available physical resources and in the number and quality of health professionals, lack of integration into secondary and tertiary care, delay in the implementation of national electronic health records, and a growing ageing population [34–36]. These challenges should be of concern and may exert chronic stress among PC providers workforce given that they have to deal with complex conditions such as mental health problems combined with economic and social issues [37]. In this regard, a number of studies have identified chronic stress and burnout syndrome as relevant issues among PC personnel in Brazil, highlighting their relation to the characteristics of the working environment [2,38–40], and prompting the need for further studies to provide better understanding of this condition, particularly as regards real prevalence, risk factors and efficacious interventions”.

- page 2, lines 79-85 and page 5, lines 214-223: the authors refer to the MBI-General Survey, that is a measure of burnout in occupational groups outside human services work (Schaufeli, Leiter, Maslach, & Jackson, 1996). Why they did not employ the MBI? Is not available in Brazil? I feel that a deeper rationale is needed to realize this choice.

We have included in the corresponding part of the manuscript that: “We used the MBI-GS, a measure of burnout in general occupational groups other than human services work in order to have comparable results with previous studies regarding the BCSQ [25,26,29,46]”. We have also recognized this as a possible limitation as follows: “we evaluated the convergence between the BCSQ and the MBI-GS as a classical measure of burnout, but this was originally developed to be used in general occupational groups. Thus, an MBI survey that is more orientated towards human services (e.g. MBI-HSS) might have allowed a more precise assessment for our study group. Nevertheless, the MBI-GS has also been used in PC health services [29], and it was used in this study to have an estimation comparable with previous research using the BCSQ”.   

Montero-Marin J, Monticelli F, Casas M, Roman A, Tomas I, Gili M, et al. Burnout syndrome among dental students: A short version of the "Burnout Clinical Subtype Questionnaire" adapted for students (BCSQ-12-SS). BMC medical education. 2011a;11(1): 103. doi: 10.1186/1472-6920-11-103.

Montero-Marin J, Zubiaga F, Cereceda M, Piva Demarzo MM, Trenc P, Garcia-Campayo J. Burnout subtypes and absence of self-compassion in primary healthcare professionals: A cross-sectional study. PLoS One. 2016;11(6): e0157499. doi: 10.1371/journal.pone.0157499

Montero-Marín J, Skapinakis P, Araya R, Gili M, García-Campayo J. Towards a brief definition of burnout syndrome by subtypes: Development of the “Burnout Clinical Subtypes Questionnaire” (BCSQ-12). Health Qual Life Out. 2011b;9(1): 74. doi: 10.1186/1477-7525-9-74

Montero-Marín J, J García-Campayo. A newer and broader definition of burnout: validation of the “Burnout Clinical Subtype Questionnaire (BCSQ-36)”. BMC Publ Health. 2010;10: 302. doi: 10.1186/1471-2458-10-302

- page 5, lines 232-238: the HADS is generally used in clinical settings with hospitalized patients. Why did the authors make this choice?

It is true that the HADS was developed for use in general medical outpatient clinics, but this scale is now widely used in clinical practice and research on both clinical and general populations. We have several examples in the literature in which the HADS was used in non-clinical samples; just to name a few: Hinz & Brähler (2011) in a German population, Crawford, Henry, Crombie & Taylor (2010) in a British population or, in South-America, Hinz, Finck, Gómez, Daig, Glaaesmer & Singer (2014) in a Colombian population. All these studies conclude that the HADS presents strong psychometric properties when used in the general population and that it is a useful instrument for assessing mental distress and identifying potential cases of anxiety or depression. In our study, we needed to assess these variables with a psychometrically strong, yet brief questionnaire -since it would be part of a battery of instruments-, and the HADS was considered the best choice. Therefore, we have included in the text that: “It was developed for use in general medical outpatient clinics, but this scale is also widely used in clinical practice and research on both clinical and general populations with strong psychometric properties [49–51]”.

Crawford, J.R., Henry, J.D., Crombie, C., & Taylor, E.P. (2010). Normative data for the HADS from a large non-clinical sample. British Journal of Clinical Psychology, 40(4), 429-434.

Hinz, A., & Brähler, E. (2011). Normative values for the Hospital Anxiety and Depression Scale (HADS) in the general German population. Journal of Psychosomatic Research, 71(2), 74-78.

Hinz, A., Finck, C., Gómez, Y., Daig, I., Glaesmer, H., & Singer, S. (2013). Anxiety and depression in the general population in Colombia: reference values of the Hospital Anxiety and Depression Scale (HADS). Social Psychiatry and Psychiatric Epidemiology, 49(1), 41-49.

- page 7, lines 299-307: I have some doubts about these multiple linear regressions. As stated by authors, the BCSQ is referred to a new model of burnout overlapping the Maslach’s theory. How can BCSQ factors and subscales be treated as independent variables in multiple linear regression analysis? Theoretical rationale for this choice should be explained by authors.

Many thanks for the opportunity to clarify this point. We have included in the text that: “The explanatory power of the BCSQ long and short versions in relation to the MBI gold standard was estimated by multiple linear regression models. It has been proposed that the burnout subtypes might be configuring three burnout risk groups which could be considered as antecedents of burnout when defined in classical terms [28]. Therefore, the total score of each MBI dimension (e.g. exhaustion, cynicism and efficacy) was considered as a dependent variable (DV), whereas the general factor scores of the BCSQ-36 – or the subscale factor scores of the BCSQ-12 – were included as independent variables (IVs)…”.

Bauernhofer K, Tanzer N, Paechter M, Papousek I, Fink A, Weiss EM. Frenetic, Under-challenged, and Worn-out: validation of the German “Burnout Clinical Subtypes Questionnaire” Student Survey and exploration of three burnout risk groups in University students. Front. Educ. 2019;4: 137. doi: 10.3389/feduc.2019.00137

- page 7, lines 321-322: could authors explain what it is meant by “Error” Reference source not found..”?

Sorry. This has been amended (it should have been “Table 1”).

-pages 7-8, lines 322-333: the sample is not well balanced for socio-demographic and occupational characteristics. This limit should be point out in the limitations paragraph (page 17, lines 618-626). Moreover, it could be useful to evaluate the role of these variables on the results by insert them in the multiple regression analysis as control variables in the first step.

We have recognized in the limitations section that: “the sample was not well balanced for socio-demographic and occupational characteristics, which posed a limitation when trying to extend the scope of results, although we controlled for them when developing multivariate regression models”. As requested by the reviewer, we have included the sociodemographic variables of age, sex, hours worked per week and job position, as a way to control for possible unbalanced characteristics in the regression models. We have created a new table (Table 7) to present these results. We have also added to the text that: “Table 7 shows the explanatory power of the general factors of the BCSQ-12 and BCSQ-36 in relation to the MBI-GS. As a first step, we included the sociodemographic and occupational variables of sex, age, hours worked per week and job position, as a way to control for possible unbalanced characteristics. This first step only explained 5% (p < .001) of exhaustion, 2% (p = .149) of cynicism, and 9% (p < .001) of efficacy. The inclusion of the BCSQ-12 factorial scores improved the explanatory power to 29% (p < .001) of exhaustion, 34% (p < .001) of cynicism, and 17% (p < .001) of efficacy, while inclusion of the BCSQ-36 latent general common factors achieved a value of 35% (p < .001) of exhaustion, 39% (p < .001) of cynicism, and 23% (p < .001) of efficacy. The standardized slopes of the burnout subtypes to explain the MBI-GS dimensions can be seen in Table 7…”.

- There are not results about retest of the two versions of BCSQ. Why did not include them in the study two on a part of the sample?

Certainly, we thought this possibility, but finally we did not include re-test measures because the sample used was going to form part of an intervention study, and thus we tried to avoid too many measurements. We have recognized this shortcoming in the limitations section.

-My recommendation for the Editor is to accept after revision.

Thank you so much for the time invested and the important contributions suggested, which have helped us to improve the quality of the manuscript.

Round 2

Reviewer 2 Report

"Many thanks for these important suggestions. We have added to the manuscript that: “At first glance, the under-challenged subtype does not seem to match the key concept of burnout, even more so if we consider burnout as a specific case of major depressive disorders. However, this profile seems to be also moderately related to exhaustion and lack of efficacy [25], creating a burnout risk group that might correspond to states of the syndrome that could have been overlooked in previous research [28]”. In the absence of other measures in previous studies, we have also included that: “… it is important to highlight the fact that the three burnout subtypes have presented adequate discriminative values regarding general negative affectivity states using the PANAS questionnaire in PC healthcare personnel, with Pearson’s correlation coefficient values ranging from r = .24 to .29 [29]”

While I do appreciate the authors efforts to improve the quality of the manuscript, the overall impression is that the authors are still in a must to explaining in a more convincing fashion, if the construct they are presenting and describing really is burnout in a classical fashion. This is even more pertinent, as a quick glance in pubmed revealed that the presented concept does not appear to be well established outside the Portuguese- and Spanish-speaking scientific community. 

"One of the most significant downsides of the classical point of view of burnout, especially regarding the 81 development of intervention strategies, is the fact that it evaluates all cases with a definition based on a scarce set of symptoms, whereas the psychosocial reality in which the syndrome develops tends to vary among cases [20,21]." this sentence is difficult to understand and to believe; actually, the three-factorial solution of the BO-questionnaires has been repeatedly confirmed. 

"....case of major depressive disorders. However, this profile seems to be also..."; specify to what "this profile" does refer.

Table 7 was helpful and nicely performed.

The authors claimed that they have modified the Discussion section and that they have shortened the Conclusion section; however, it does not seem so. 

Author Response

While I do appreciate the authors efforts to improve the quality of the manuscript, the overall impression is that the authors are still in a must to explaining in a more convincing fashion, if the construct they are presenting and describing really is burnout in a classical fashion. This is even more pertinent, as a quick glance in pubmed revealed that the presented concept does not appear to be well established outside the Portuguese- and Spanish-speaking scientific community.

Thank you for giving us the opportunity of explaining this in more detail. We have added to the manuscript that: “as an alternative to the traditional definition, distinct burnout subtypes (e.g. frenetic, under-challenged, and worn-out) that point to various groups at risk of burnout syndrome have been proposed from Farber's seminal clinical work [20,22,25–30]. The definitions of these burnout profiles are the result of a unifying methodological process, originating with this author’s clinical approach and based on a purely phenomenological description of cases under psychological treatment, which was subjected to a qualitative analysis of content prior to the subsequent validation, in psychometric terms, of the ‘Burnout Clinical Subtypes Questionnaire’ (BCSQ-36/BCSQ-12) [31,32]. The BCSQ is a recently developed tool that permits the measurement of the referred three distinct burnout profiles. It allows the identification of risk groups rather than cases of burnout in a classical sense, and thus highlights the complex and multifaceted nature of the syndrome by facilitating a more person-orientated approach to burnout [33]. This framework is focused on psychological processes that are relevant when intervening on the specific characteristics of each particular case [34]. The BCSQ has already been previously used in different languages and in several countries, such as Spain [31], Iran [35], Sri Lanka [36], India [37], and Austria [38], among others, and it is now being translated and validated in many others”.

"One of the most significant downsides of the classical point of view of burnout, especially regarding the development of intervention strategies, is the fact that it evaluates all cases with a definition based on a scarce set of symptoms, whereas the psychosocial reality in which the syndrome develops tends to vary among cases [20,21]." this sentence is difficult to understand and to believe; actually, the three-factorial solution of the BO-questionnaires has been repeatedly confirmed.

Thank you for allowing us to clarify this very important point. We regret we were unable to explain it with enough detail previously. We are not referring here to factorial solutions or structures. To make this point clear, we have added to the text pointed out by the reviewer that: “In other words, burnout syndrome has usually been described as a uniform construct in all individuals, with a similar aetiology and group of symptoms [22]; however, experience in the treatment of this complex psychosocial entity suggests the need to identify different routes in the development of the syndrome in order to adjust for more effective lines of therapeutic action [21]”.

"....case of major depressive disorders. However, this profile seems to be also..."; specify to what "this profile" does refer.

We have clarified that “However, the previously indicated under-challenged profile…” in line with which was proposed by the reviewer in the previous round.

Table 7 was helpful and nicely performed.

Thank you very much.

The authors claimed that they have modified the Discussion section and that they have shortened the Conclusion section; however, it does not seem so.

In the first round, we reduced the length of conclusions from 680 words to 600 words. Now, we have further shortened it to 480 words. We think any additional reductions will affect the content of this section. We hope the reviewer is satisfied with this new version.  

Reviewer 3 Report

The authors addressed satisfactorily all my comments. Therefore, I endorse the publication of this manuscript in its current form.

Author Response

Thank you very much.